# Continuous Control of LLM Text Generation via Probabilistic Prompting

## Abstract

The scarcity of high-quality labeled data remains a critical bottleneck in natural language processing, while existing synthetic data generation approaches using Large Language Models (LLMs) rely on rigid categorical conditioning that produces polarized and unrealistic text. We introduce Probabilistic Prompting, a framework that achieves fine-grained control over LLM text generation by conditioning on continuous probability vectors rather than discrete categorical instructions. To realize this framework, we propose SoftGen, a zero-shot method implementing a three-stage pipeline: (1) sampling probability vectors from tailored distributions, (2) generating text conditioned on probabilistic prompts, and (3) self-verification to ensure high fidelity. Through comprehensive evaluation on five text classification benchmarks, we demonstrate three key contributions. First, we provide rigorous analysis of generation fidelity, revealing that LLMs can faithfully follow probabilistic instructions and uncovering systematic relationships between label entropy and generation quality that vary by task dimensionality. Second, we show substantial downstream utility: models trained on our synthetic data achieve improved accuracy and calibration compared to traditional categorical approaches. Third, we establish theoretical foundations grounded in the Maximum Entropy Principle, including formal definitions of Generator Calibration and mathematical proofs connecting prompt entropy to output diversity. Our work demonstrates that preserving continuous probability structure in synthetic data generation provides richer supervisory signals and enables more realistic, diverse datasets better reflecting the continuous nature of semantic properties in natural language.

## 1 Introduction

The scarcity of high-quality labeled data remains a critical bottleneck in natural language processing (NLP) (Bansal et al., 2022). While Large Language Models (LLMs) have emerged as a powerful tool for remedying this through synthetic data generation, a fundamental challenge remains: achieving the precise, fine-grained control necessary to produce realistic and diverse datasets (Li et al., 2023). Prevailing approaches often rely on categorical conditioning: prompting an LLM with a discrete instruction such as "positive" or "negative" to generate text aligned with that category (Ye et al., 2022a). This imposes rigid boundaries on inherently continuous linguistic phenomena, like sentiment, emotion, or stylistic tone, often yielding polarized and unrealistic text (Cowen & Keltner, 2017).

To overcome these limitations and enable a higher degree of controllability, we introduce and analyze a new framework for LLM control, which we term **Probabilistic Prompting**[1]. The core principle is to move beyond categorical instructions (e.g., "generate a positive movie review") and instead condition the generative process on a continuous probability vector, or "soft label" (e.g., "generate a review that is 70% positive"). This provides a more expressive control mechanism over the model's output. To realize this framework, we propose SoftGen, a zero-shot method that implements probabilistic prompting through a principled three-stage pipeline designed to achieve high-fidelity controllable generation.

Our work makes three primary scientific and empirical contributions. First, we provide a deep analysis of **generation fidelity**, formally demonstrating that LLMs can faithfully adhere to probabilistic instructions and uncovering a novel systematic relationship between label entropy and generation

---

[1] Our "Probabilistic Prompting" should not be confused with "Probabilistic Prompt Learning" methods in computer vision, which learn prompt embeddings rather than conditioning generation on explicit probability vectors expressed in natural language.

fidelity. Second, we demonstrate the **downstream utility** of this high-fidelity data, showing that models trained with our method are not only more accurate but also better-calibrated. Finally, we develop a **rigorous theoretical framework** for controllable text generation grounded in the Maximum Entropy Principle (Jaynes, 1957), including formal definitions of Generator Calibration, mathematical proofs establishing the relationship between prompt entropy and output diversity, and theoretical performance bounds that validate our empirical findings.

By introducing Probabilistic Prompting and demonstrating its effectiveness through SoftGen, we establish new theoretical foundations for controllable generation that bridge continuous semantic properties with discrete classification tasks, opening avenues for more principled synthetic data creation.

## 2 RELATED WORK

**Synthetic Data Generation with LLMs**    The use of LLMs to generate synthetic data offers a scalable alternative to manual annotation (Lu et al., 2025). The standard paradigm, established by foundational work like ZeroGen (Ye et al., 2022a), relies on categorical conditioning, where the LLM is prompted with a discrete class label to generate text (Li et al., 2023; Yu et al., 2023). However, empirical studies have consistently shown that this approach has significant limitations, often producing datasets that are polarized, lack diversity, and generalize poorly, especially for tasks involving subjective or nuanced judgments (Yu et al., 2023; Møller et al., 2024).

To address these shortcomings, a second generation of more sophisticated methods has emerged. These approaches, such as the iterative refinement pipelines of ProGen (Ye et al., 2022b) and GOLD (Gholami et al., 2024) or the post-hoc re-weighting of SunGen (Gao et al., 2023), attempt to improve upon an initial hard-labeled dataset. Our work explores a different and complementary direction. Instead of refining a flawed initial dataset, we aim to improve the quality of the data at its source by fundamentally changing the nature of the initial conditioning signal itself, moving from discrete categories to a continuous and probabilistic instruction.

**Soft Labels: From Post-Hoc Annotation to Generative Instruction**    The benefits of training on soft labels are well-established, with theoretical roots in regularization techniques like label smoothing (Müller et al., 2019) and knowledge distillation (Hinton et al., 2015). Prior methods, from Mixup in computer vision (Zhang et al., 2017) to GPT3Mix in NLP (Yoo et al., 2021), leverage this principle by creating soft targets in a post-hoc step, after the primary data has been generated or interpolated. However, in these approaches, the soft label is a passive annotation and does not influence the generative process itself. Our work marks a fundamental departure. Instead of assigning soft labels after the fact, our framework directly embeds them into the prompt to guide the generation from the outset. This invertible conditioning from label to text, rather than text to label, allows us to not only leverage the known regularizing benefits of soft targets but also to generate text that is inherently more nuanced and faithful to the specified probabilistic instruction.

## 3 METHODOLOGY

### 3.1 THE PROBABILISTIC PROMPTING FRAMEWORK

We introduce **Probabilistic Prompting**, a general framework for achieving fine-grained control over the semantic properties of text generated by LLMs. The core principle is to move beyond the limitations of discrete, categorical conditioning and instead use a continuous probability vector, or "soft label", as the primary instruction for the generative process.

Formally, the proposed framework is defined by three key components $(\mathcal{S}, \mathcal{P}, \pi)$, where:

1. $\mathcal{S}$ is a continuous semantic space. For a given task with $K$ semantic attributes (e.g., classes, stylistic dimensions), the control space $\mathcal{S}$ is defined as the probability simplex $\Delta^{K-1}$. Each point $\boldsymbol{p} \in \mathcal{S}$ represents a unique and nuanced blend of the attributes.

2. $\mathcal{P}$ is a probability distribution defined over the semantic space $\mathcal{S}$. The choice of this distribution is a flexible component that allows for tailoring the generation process. A specific soft label $\boldsymbol{p}$ is sampled from this distribution: $\boldsymbol{p} \sim \mathcal{P}(\cdot)$.

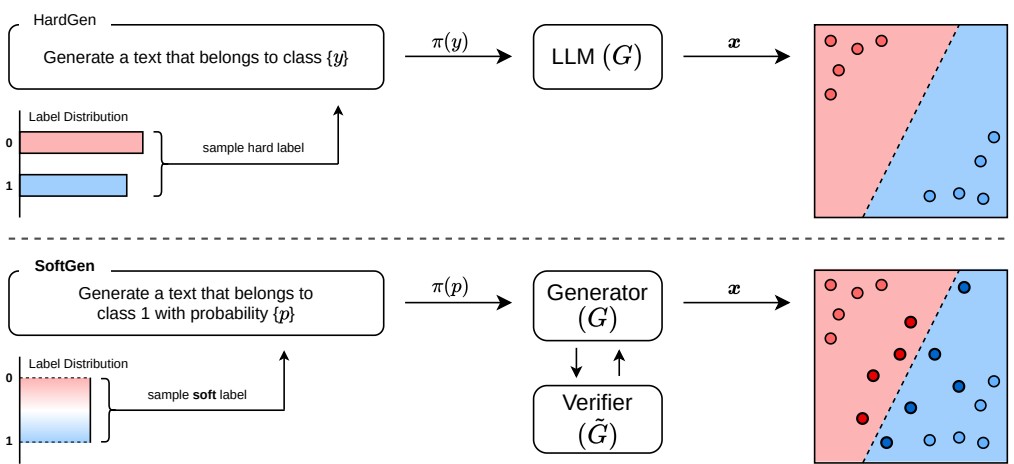

Figure 1: Comparison of HardGen baseline (top) using discrete class labels versus SoftGen (bottom) using continuous probability distributions with self-verification.

3. $\pi$ is a prompt construction function. This is a mapping $\pi : \mathcal{S} \to \mathcal{T}_{\text{prompt}}$ that transforms a sampled vector $\boldsymbol{p}$ from the semantic space into a natural language instruction $\pi(\boldsymbol{p})$ in the space of possible prompts, $\mathcal{T}_{\text{prompt}}$.

The final generative process is thus defined as sampling a text $\boldsymbol{x}$ from an LLM $G$ conditioned on the constructed prompt: $\boldsymbol{x} \sim G(\cdot | \pi(\boldsymbol{p}))$, where $\boldsymbol{p} \sim \mathcal{P}(\cdot)$.

## 3.2 PROBLEM SETUP AND BASELINE

While the Probabilistic Prompting framework is general, in this work, we focus specifically on its application to text classification, where the $K$ semantic attributes represent class memberships. Our objective is to generate a dataset $\mathcal{D} = \{(\boldsymbol{x}, \boldsymbol{p})\}$, where each text $\boldsymbol{x}$ is generated to align with a target class distribution, specified by a soft-label vector $\boldsymbol{p} \in \Delta^{K-1}$.

As a baseline, we implement the standard paradigm of hard-label conditioning for zero-shot text generation, a technique established by prior work such as ZeroGen (Ye et al., 2022a). We term our implementation of this baseline **HardGen** to create a clear conceptual contrast with our work. In this method, a discrete class label $y \in \{1, ..., K\}$ is sampled from a categorical distribution and then embedded in a simple prompt (e.g., "Generate a positive review").

## 3.3 SOFTGEN: A METHOD FOR PROBABILISTIC PROMPTING

To realize the paradigm of Probabilistic Prompting, we propose **SoftGen**, a zero-shot method for generating soft-labeled text data. SoftGen instantiates the abstract framework $(\mathcal{S}, \mathcal{P}, \pi)$ with a concrete three-stage pipeline designed to ensure both flexibility and high data fidelity.

**(1) Label Sampling** Our method begins by strategically sampling a probability vector $\boldsymbol{p}$ from a chosen distribution, $\mathcal{P}(\cdot)$. A key strength of our approach is the flexibility in this choice, which allows practitioners to tailor the data generation process to their specific needs. For instance, if no prior knowledge of the true label distribution is available, a uniform distribution can be used as an unbiased default. Conversely, domain knowledge can be used to employ a more specific distribution to emphasize underrepresented regions of the label space. For binary classification ($K = 2$), any user-specified distribution over probability parameter $p \in [0, 1]$ can be used, such as the Beta$(\alpha, \beta)$ distribution, yielding probability vectors $\boldsymbol{p} = [1 - p, p]$. For multiclass tasks ($K > 2$), a distribution over the simplex, like Dirichlet$(\boldsymbol{\alpha})$, is appropriate.

**(2) Text Generation** Next, the sampled soft label vector $\boldsymbol{p}$ is encoded into a natural language prompt $\pi(\boldsymbol{p})$ that instructs the LLM generator $G$ on the desired semantic properties of the text. (e.g.,

$\pi([0.3, 0.7]) =$"Generate a review expressing approximately 70% positivity"). The text $\boldsymbol{x}$ is then generated by sampling from the LLM conditioned on this prompt:

$$\boldsymbol{x} \sim G(\cdot | \pi(\boldsymbol{p}))$$

**(3) Self-Verification** Finally, to promote high data fidelity, the generated text $\boldsymbol{x}$ undergoes a self-verification stage (Weng et al., 2022; Gero et al., 2023; Pan et al., 2024). This process acts as a form of *peer review*: an independent instance of the same LLM takes the role of a *verifier*, $\tilde{G}$, to predict a post-hoc soft label $\tilde{\boldsymbol{p}}$ without knowledge of the original prompt or the conditioned soft label $\boldsymbol{p}$:

$$\tilde{\boldsymbol{p}} = \tilde{G}(\boldsymbol{x})$$

We then apply a discrepancy filter, where a sample $(\boldsymbol{x}, \boldsymbol{p})$ is accepted into final dataset only if the $\ell_1$ distance between the intended and verified labels is below a predefined threshold, $\tau$. Formally, the final dataset is defined as:

$$\mathcal{D} = \{(\boldsymbol{x}, \boldsymbol{p}) \mid \|\boldsymbol{p} - \tilde{\boldsymbol{p}}\|_1 \leq \tau\}$$

This verification step filters out low-fidelity examples where the generator failed to produce text that aligns with the intended semantic properties. For a formal step-by-step description of the SoftGen pipeline, please see Algorithm 1 in the Appendix A.

## 4 EXPERIMENTS

In this section, we present a comprehensive empirical evaluation of our SoftGen approach. Our experiments are designed to answer two key questions: (*i*) How faithfully does the generated text align with the probabilistic instructions in the prompt? (*ii*) How useful is the generated high-fidelity data for training robust and well-calibrated downstream classifiers?

### 4.1 GENERAL SETUP

**Datasets.** Our evaluation is conducted on five public text classification benchmarks: **IMDb** (Maas et al., 2011) and **SST** (Socher et al., 2013) for binary sentiment analysis; **SUBJ** (Pang & Lee, 2004) for binary subjectivity detection; **Emotion** (Saravia et al., 2018) for 6-way emotion classification; and **Yahoo! Answers** (Zhang et al., 2015) for 10-way question category classification. These datasets cover a range of domains, text lengths, and label complexities. A detailed overview of these benchmarks, including summary statistics, is provided in Appendix B.

**Generation Details** For each benchmark, we generate a synthetic dataset $\mathcal{D}$ whose size matches that of the original human-annotated training set. We use **gemini-2.0-flash** as the LLM for both generation ($G$) and self-verification ($\tilde{G}$) tasks. To foster the generation of diverse and high-quality text, all prompts use a sampling temperature of 1.0 and include a zero-shot Chain-of-Thought (CoT) instruction (i.e., "think step-by-step") (Wei et al., 2022). For our SoftGen method, we employ a principled sampling strategy motivated by our fidelity analysis in Section 4.2: unless otherwise specified, we sample labels from a U-shaped Beta$(0.5, 0.5)$ distribution for binary tasks, and a symmetric Dirichlet($\boldsymbol{0.5}$) distribution for multiclass tasks. To ensure samples maintain high-fidelity, we set the verification threshold $\tau = \varepsilon \times K$, where $\varepsilon$ represents the maximum acceptable average absolute error per component. For our experiments, we use $\varepsilon = 0.05$, corresponding to $5\%$ average error per component, yielding $\tau = 0.05K$. We provide the full templates for all datasets in Appendix C.

### 4.2 ANALYSIS OF GENERATION FIDELITY

A core premise of our Probabilistic Prompting framework is the ability to reliably control the semantic properties of the generated text. But how faithfully does the generated text actually align with the probabilistic instructions in the prompt? We provide a multi-faceted quantitative analysis to answer this question by breaking it down into three distinct sub-questions.

**Fidelity Evaluation Setup** Prior to any analysis, we require a proxy for the ground-truth semantic properties (i.e., soft labels) of the generated text. Acknowledging the potential limitations of any

Table 1: Alignment between input prompt vectors $p$ and judge predictions $p_{\text{judge}}$, measured by Pearson correlation for binary tasks and average cosine similarity for multiclass tasks. Results averaged over five trials demonstrate strong directional alignment across both evaluation approaches.

| Task Type | Dataset | Gold-Trained | LLM-as-a-Judge | | |
|---|---|---|---|---|---|
| | | Judge | gpt-5 | claude-sonnet-4 | gemini-2.5-pro |
| Binary | IMDb | 0.8856 | 0.9955 | 0.9891 | 0.9927 |
| | SST | 0.8712 | 0.9896 | 0.9924 | 0.9916 |
| | SUBJ | 0.7339 | 0.9558 | 0.9443 | 0.9867 |
| Multiclass | Emotion | 0.7813 | 0.9115 | 0.9215 | 0.9077 |
| | Yahoo! Answers | 0.7551 | 0.8432 | 0.8558 | 0.8679 |

single evaluation method, we employ two distinct types of judge models to generate a proxy ground-truth soft label $p_{\text{judge}} \in \Delta^{K-1}$. Our primary proxy is (*i*) the LLM-as-a-Judge method (Zheng et al., 2023) in order to measure a form of internal consistency. It answers the question: "Does the generated text align with the general understanding of language shared by other powerful language models?" We use a diverse panel of three independent models for this role: **gpt-5**, **claude-sonnet-4**, and **gemini-2.5-pro**. To complement this and ensure the robustness of our findings, we use a (*ii*) **Gold-Trained Judge**, a DistilBERT classifier finetuned on 100% of the real human-annotated training data measuring domain-specific fidelity. It answers the question: "Does the generated text align with the specific, and potentially idiosyncratic, labeling patterns of the original human annotators for this particular dataset"? As shown below, the fact that these two distinct validation approaches yield a consistent conclusion provides a robust and reliable foundation of our fidelity analysis.

**Question 1: Does the generator follow the correct direction?** First, we test for basic alignment by measuring the correlation (binary) or average cosine similarity (multiclass) between the prompt vectors $p$ and the judge-perceived vectors $p_{\text{judge}}$, with the results summarized in Table 1. The consistently high alignment scores with the LLM-as-a-Judge panel confirm that our SoftGen control mechanism is working reliably on its own terms. Crucially, the strong positive correlation with the Gold-Trained Judge provides a yet more powerful piece of evidence that the generation fidelity is a genuine phenomenon grounded in the specific domain of each dataset, not merely an artifact of LLMs evaluating other LLMs. The fact that these two distinct approaches—one measuring general consistency, the other measuring domain-specific fidelity—yield the same conclusion provides strong evidence that the generator correctly interprets and follows the directional intent of the continuous prompt instructions.

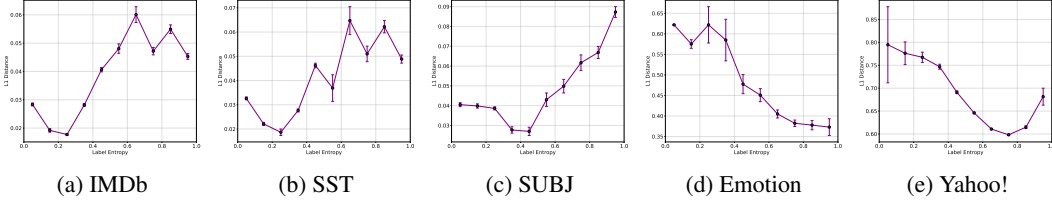

| (a) IMDb | (b) SST | (c) SUBJ | (d) Emotion | (e) Yahoo! |

Figure 2: Generation error magnitude as a function of prompt entropy across datasets (judge: gpt-5). Each plot shows the $\ell_1$ distance between prompt vectors $p$ and judge predictions $p_{\text{judge}}$ (y-axis) versus prompt entropy (x-axis). Binary tasks (a-c) exhibit positive correlations between entropy and error, while multiclass tasks (d-e) show negative correlations, demonstrating systematic and predictable error patterns.

**Question 2: How precise is the generator, and is its error predictable?** Moving beyond directional alignment, we analyze the magnitude of generation error across the label spectrum. Figure 2 plots the $\ell_1$ distance between prompt and judge vectors, $|p - p_{\text{judge}}|$, revealing systematic relationships between label entropy and generation fidelity that vary by task dimensionality. Binary classification tasks (a-c) exhibit positive correlations between label entropy and generation error, indicating that ambiguous prompts near decision boundaries are more challenging to execute than

clear categorical instructions. Conversely, both multiclass tasks (d-e) show negative correlations where higher entropy prompts yield better fidelity. We hypothesize this reversal reflects the generator's conservative bias toward producing balanced, moderate text (Kirk et al., 2023): low-entropy prompts requesting extreme single-category dominance conflict with this natural tendency, while high-entropy prompts specifying mixed content align with the generator's intrinsic preference for realistic and nuanced expressions. These systematic error patterns demonstrate that generation error follows predictable principles rather than random variation, with the specific relationships varying across task types and semantic domains.

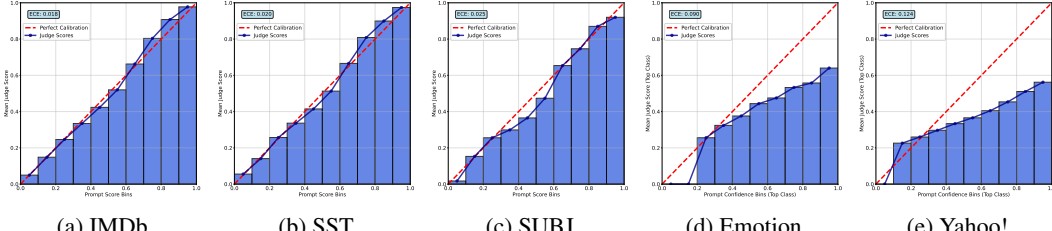

(a) IMDb     (b) SST     (c) SUBJ     (d) Emotion     (e) Yahoo!

Figure 3: Generator calibration analysis showing the relationship between prompted confidence levels (x-axis) and judge-perceived confidence (y-axis) using gpt-5 as judge. The diagonal line represents perfect calibration where prompted and perceived confidences match exactly. For multiclass tasks, we analyze the maximum probability class. Proximity to the diagonal indicates good calibration, while systematic deviations reveal directional biases in generation.

**Question 3: Is the generator systematically biased?** Finally, we analyze the directional bias of the generation error. A generator could be precise but systematically biased (e.g., always generating text that is 5% more positive than requested). To investigate this, we introduce a Generator Calibration Plot, a reliability diagram adapted for controllable generation. We bin the generated samples by their prompt score (the x-axis) and plot the average judge score for each bin (the y-axis). As shown in Figure 3, our method demonstrates good calibration with low ECE, but reveals a systematic underconfidence bias where the generator consistently produces more moderate text than explicitly requested. This underconfidence bias helps explain the contrasting entropy-error relationships observed in Question 2: while binary tasks favor low-entropy prompts that align with clear categorical distinctions, multiclass tasks benefit from high-entropy prompts that match the generator's natural tendency toward balanced, moderate outputs across multiple semantic dimensions.

Together, the answers to these three questions provide comprehensive evidence for the systematic controllability and predictable reliability offered by our SoftGen framework, revealing principled strategies for optimization across different task types.

## 4.3 DOWNSTREAM CLASSIFICATION PERFORMANCE

**Setup** Having established generation fidelity, we evaluate downstream utility by comparing Soft-Gen against rigorous baselines. **Gold** (trained on original data) serves as an upper bound. Zero-shot baselines include **HardGen** (standard hard-label conditioning), **HardGen+EDA** (with synonym replacement/insertion augmentations), and **HardGen+PostHocSoft** (hard generation with post-hoc soft labels via our verification procedure). We test **SoftGen+HardTarget** (discretized labels) and **SoftGen+SoftTarget** (direct soft label training). All methods use DistilBERT with LoRA, evaluated on original test sets following standard protocols (Ye et al., 2022a;b). Implementation details are in Appendix E.

**Results** The results in Table 2 demonstrate two key benefits of our framework. First, the quality of generated text itself provides significant performance advantages: across all datasets, SoftGen+HardTarget robustly outperforms hard-label baselines (HardGen, HardGen+EDA, Hard-Gen+PostHocSoft). For instance, on IMDb, SoftGen+HardTarget achieves 83.3% accuracy versus the strongest baseline's 76.8%, underscoring the benefit of training on semantically diverse datasets with borderline examples. Second, training directly on soft labels (SoftGen+SoftTarget) consistently improves both accuracy and calibration across most datasets, with the notable exception of Yahoo!

Table 2: Downstream classification performance showing accuracy/F1 (↑) and Expected Calibration Error (ECE, ↓). The best performance among zero-shot methods for each metric is in **bold**, and second best underlined. Standard errors across five trials in parentheses.

| Method | IMDb | | SST | | SUBJ | | Emotion | | Yahoo! | |
|---|---|---|---|---|---|---|---|---|---|---|
| | Acc. | ECE | Acc. | ECE | Acc. | ECE | F1 | ECE | Acc. | ECE |
| Gold | 92.1 (0.1) | 4.1 (0.1) | 83.4 (1.0) | 2.7 (0.5) | 96.5 (0.1) | 1.01 (0.1) | 89.7 (0.1) | 3.17 (0.2) | 71.9 (0.1) | 1.4 (0.2) |
| *(Baselines)* | | | | | | | | | | |
| HardGen | 75.3 (1.4) | 21.8 (0.9) | 69.3 (1.2) | 27.4 (0.8) | 63.7 (0.5) | 29.9 (0.3) | 35.2 (1.9) | 33.9 (0.7) | 54.2 (1.3) | 15.2 (0.9) |
| HardGen+EDA | 76.8 (1.4) | 21.5 (1.0) | 70.8 (1.4) | 27.3 (0.9) | 66.2 (0.8) | 30.3 (0.5) | 38.2 (2.1) | 34.2 (0.8) | 55.4 (1.5) | 15.0 (1.0) |
| HardGen+PostHocSoft | 75.4 (0.9) | 21.2 (0.8) | 69.8 (1.1) | 27.4 (0.7) | 63.7 (0.5) | 29.9 (0.3) | 35.5 (0.4) | 31.6 (0.4) | 54.2 (1.1) | 15.3 (0.8) |
| *(Ours)* | | | | | | | | | | |
| SoftGen+HardTarget | 83.3 (0.2) | 13.8 (0.3) | 78.2 (0.7) | 11.7 (1.1) | 75.8 (0.4) | 14.6 (0.4) | 42.8 (0.3) | 11.9 (0.5) | 55.3 (0.1) | **5.12** (0.2) |
| SoftGen+SoftTarget | **84.8** (0.1) | **9.1** (0.2) | **79.0** (0.3) | **10.1** (0.4) | **79.5** (0.2) | **9.8** (0.3) | **47.1** (0.2) | **10.2** (0.2) | **61.2** (1.5) | 14.73 (1.6) |

where calibration degrades. This demonstrates that our framework produces not just higher-quality text, but also richer supervisory signals, as training with soft labels is mathematically equivalent to adding a data-dependent regularizer (Hinton et al., 2015; Müller et al., 2019).

## 4.4 ABLATION STUDIES

Ablation studies validate our key design choices (Table 3). Chain-of-Thought prompting provides consistent 2-8 point improvements across all datasets, while self-verification shows substantial benefits for most tasks: large gains for SUBJ (+6.2), Yahoo! (+5.7), and Emotion (+3.4 F1), but minimal impact for IMDb and SST. This suggests verification is most valuable for tasks with greater semantic complexity or ambiguity. Additional ablations on sampling distribution choices and soft vs. hard target training are provided in Appendix D, confirming that our design decisions contribute meaningfully to the framework's effectiveness.

Table 3: Ablation study on Chain-of-Thought prompting and self-verification components. Primary performance metrics shown with standard errors from five trials.

| Method | IMDb (Acc. ↑) | SST (Acc. ↑) | SUBJ (Acc. ↑) | Emotion (F1 ↑) | Yahoo! (Acc. ↑) |
|---|---|---|---|---|---|
| SoftGen+SoftTarget (full method) | 84.8 (±0.1) | 79.0 (±0.3) | 79.5 (±0.2) | 47.1 (±0.2) | 61.2 (±1.5) |
| *Ablations* | | | | | |
| – w/o CoT | 76.6 (±1.9) | 74.2 (±2.5) | 76.9 (±2.2) | 45.9 (±0.9) | 53.1 (±1.9) |
| – w/o Verification | 84.1 (±0.3) | 77.8 (±1.3) | 73.3 (±3.1) | 43.7 (±1.1) | 55.5 (± 0.9) |

Table 4: Inter- and intra-class cosine similarity of generated text embeddings. Lower intra-class values indicate higher within-class diversity. Inter-class interpretation discussed in text.

| Method | IMDb | | SST | | SUBJ | | Emotion | | Yahoo! | |
|---|---|---|---|---|---|---|---|---|---|---|
| | Inter | Intra (↓) | Inter | Intra (↓) | Inter | Intra (↓) | Inter | Intra (↓) | Inter | Intra (↓) |
| Gold | 0.3555 | 0.3950 | 0.2531 | 0.2705 | 0.1991 | 0.2520 | 0.2665 | 0.2874 | 0.1587 | 0.2293 |
| HardGen | 0.5032 | 0.8917 | 0.3494 | 0.7225 | 0.0391 | 0.3481 | 0.2724 | 0.6565 | 0.1455 | 0.5514 |
| SoftGen | 0.6246 | 0.7188 | 0.4841 | 0.6739 | 0.1851 | 0.2811 | 0.4297 | 0.4998 | 0.2202 | 0.2699 |

## 4.5 ANALYSIS OF TEXT DIVERSITY

Beyond downstream performance, we evaluate text diversity via pairwise cosine similarity using OpenAI's text-embedding-3-small model. As shown in Table 4, SoftGen consistently produces lower intra-class similarity than HardGen, indicating greater within-class variety. Compared to Gold standard, SoftGen achieves comparable diversity levels across datasets. Meanwhile, the inter-class similarity patterns require careful interpretation. HardGen's discrete sampling creates polarized

data with large semantic gaps, while SoftGen exhibits higher inter-class similarity consistent with generating more examples near decision boundaries. This aligns with our goal of creating nuanced, borderline examples that traditional approaches cannot produce.

## 5 THEORETICAL ANALYSIS

To provide theoretical grounding for our empirical findings, we establish the fundamental relationship between prompt entropy and output diversity. We build this systematically, starting with the theoretical foundation and precise mathematical definitions.

### 5.1 THEORETICAL FOUNDATION

Our approach builds on the **Maximum Entropy Principle** (MaxEnt): *when making inferences from partial information, choose the probability distribution that maximizes entropy subject to known constraints* (Jaynes, 1957). For controllable text generation, this means: seek the most diverse text distribution that satisfies the semantic constraints encoded in the prompt.

This principle provides the theoretical foundation for controlled text generation: we want maximum diversity (highest entropy) while maintaining perfect calibration with semantic constraints. This is the "least biased" approach, avoiding artificial restrictions beyond those required by the prompt.

### 5.2 DEFINITIONS

**Definition 1** (Generator Calibration). *Let $\mathcal{X}$ denote a finite space of possible texts, and let $\boldsymbol{p}^*$ : $\mathcal{X} \to \Delta^{K-1}$ be a labeling function that assigns each text a distribution over $K$ categories. A generative model $G$ with prompt construction function $\pi$ is **perfectly calibrated** if, for any prompt vector $\boldsymbol{p} \in \Delta^{K-1}$, the expected label distribution of generated texts equals the prompt:*

$$\mathbb{E}_{\boldsymbol{x} \sim G(\cdot|\pi(\boldsymbol{p}))}[\boldsymbol{p}^*(\boldsymbol{x})] = \boldsymbol{p} \tag{1}$$

Perfect calibration means that if we prompt for "70% positive, 30% negative" sentiment via $\pi(\boldsymbol{p})$, the generated texts will have exactly those average sentiment scores. This formalizes "prompt following" while acknowledging that the generator only sees the natural language prompt $\pi(\boldsymbol{p})$, not the probability vector $\boldsymbol{p}$ directly.

**Definition 2** (Idealized Rational Generator). *An **idealized rational generator** $G^*$ with prompt construction function $\pi$ implements the Maximum Entropy Principle: given a prompt vector $\boldsymbol{p}$, it selects the distribution over texts that maximizes entropy $H$ subject to perfect calibration:*

$$P^*(\cdot|\boldsymbol{p}) = \underset{P \in \mathcal{P}(\mathcal{X})}{\arg\max} \quad H(P) \quad s.t. \quad \mathbb{E}_{\boldsymbol{x} \sim P}[\boldsymbol{p}^*(\boldsymbol{x})] = \boldsymbol{p} \tag{2}$$

*The idealized rational generator $G^*$ samples from this distribution: $\boldsymbol{x} \sim G^*(\cdot|\pi(\boldsymbol{p})) = \boldsymbol{x} \sim P^*(\cdot|\boldsymbol{p})$.*

This formalizes the ideal controllable generator: it produces the most diverse possible text distribution while perfectly satisfying the semantic constraints. The MaxEnt principle ensures this is the least biased solution.

**Definition 3** (Maximum Achievable Entropy). *For a given prompt vector $\boldsymbol{p}$, we define $H^*(\boldsymbol{p})$ as the **maximum achievable entropy**:*

$$H^*(\boldsymbol{p}) = \max_{P \in \mathcal{P}(\mathcal{X})} \{H(P) : \mathbb{E}_{\boldsymbol{x} \sim P}[\boldsymbol{p}^*(\boldsymbol{x})] = \boldsymbol{p}\} \tag{3}$$

*This represents the entropy of the optimal distribution produced by the idealized rational generator for prompt $\boldsymbol{p}$.*

The optimal solution to Equation (2) has **exponential family form**:

$$P^*(\boldsymbol{x}|\boldsymbol{p}) = \frac{\exp(\boldsymbol{\lambda}^*(\boldsymbol{p})^T \boldsymbol{p}^*(\boldsymbol{x}))}{Z(\boldsymbol{\lambda}^*(\boldsymbol{p}))} \tag{4}$$

where $Z(\boldsymbol{\lambda}) = \sum_{\boldsymbol{x}} \exp(\boldsymbol{\lambda}^T \boldsymbol{p}^*(\boldsymbol{x}))$ is the partition function, and $\boldsymbol{\lambda}^*(\boldsymbol{p})$ is the unique Lagrange multiplier satisfying the constraint $\mathbb{E}_{P^*}[\boldsymbol{p}^*(\boldsymbol{x})] = \boldsymbol{p}$. The derivation appears in Appendix F.

## 5.3 ASSUMPTIONS

To establish the entropy correspondence rigorously, we need precise assumptions about the text space structure:

**Assumption 1** (Balanced Text Space). *The text space contains equal numbers of texts favoring each category:* $|\{\boldsymbol{x} : argmax(\boldsymbol{p}^*(\boldsymbol{x})) = i\}| = N$ *for all categories* $i$.

**Assumption 2** (Symmetric Label Structure). *For binary classification* $(K = 2)$, *the text space exhibits reflection symmetry: for every text* $\boldsymbol{x}$ *with label* $\boldsymbol{p}^*(\boldsymbol{x}) = [p, 1 - p]$, *there exists a corresponding text* $\boldsymbol{x}'$ *with the reflected label* $\boldsymbol{p}^*(\boldsymbol{x}') = [1 - p, p]$. *For multi-class cases, this generalizes to appropriate permutation symmetries around the uniform distribution.*

**Assumption 3** (Smooth Constraint Response). *The optimal Lagrange multiplier* $\boldsymbol{\lambda}^*(\boldsymbol{p})$ *varies smoothly with the prompt* $\boldsymbol{p}$, *and the constraint set* $\{\mathbb{E}_P[\boldsymbol{p}^*(\boldsymbol{x})] : P \in \Delta(\mathcal{X})\}$ *contains* $\boldsymbol{p}$ *in its interior.*

**Assumption 4** (Perfect Prompt Interpretation). *The prompt construction function* $\pi : \Delta^{K-1} \to \mathcal{L}$ *(where* $\mathcal{L}$ *is the space of natural language prompts) acts as a perfect information channel: the generator can perfectly recover the intended constraint* $\mathbb{E}_{\boldsymbol{x} \sim P}[\boldsymbol{p}^*(\boldsymbol{x})] = \boldsymbol{p}$ *from the natural language prompt* $\pi(\boldsymbol{p})$. *The practical implications and limitations of this assumption are discussed in Appendix F.3.*

## 5.4 MAIN THEOREM: BINARY ENTROPY-DIVERSITY CORRESPONDENCE

**Theorem 1** (Binary Entropy-Diversity Correspondence). *Consider the binary classification case* $(K = 2)$ *under Assumptions 1–4. For prompts* $\boldsymbol{p}_1 = [p_1, 1 - p_1]$ *and* $\boldsymbol{p}_2 = [p_2, 1 - p_2]$ *with* $p_1, p_2 \in (0, 1)$:

*If* $H(\boldsymbol{p}_1) < H(\boldsymbol{p}_2)$, *then* $H^*(\boldsymbol{p}_1) < H^*(\boldsymbol{p}_2)$.

*Equivalently: if* $|p_1 - 0.5| > |p_2 - 0.5|$ *(i.e.,* $\boldsymbol{p}_1$ *is more extreme than* $\boldsymbol{p}_2$*), then* $H^*(\boldsymbol{p}_1) < H^*(\boldsymbol{p}_2)$.

**Interpretation**: More balanced binary prompts (closer to $[0.5, 0.5]$) yield higher entropy output distributions than extreme prompts (closer to $[1, 0]$ or $[0, 1]$). This establishes that **prompt entropy directly controls output diversity** in optimal MaxEnt-based controllable generation.

*Proof Sketch.* The proof exploits the reflection symmetry of the binary text space. The symmetry assumption ensures that constraints $\mathbb{E}[p_1^*(\boldsymbol{x})] = p$ and $\mathbb{E}[p_1^*(\boldsymbol{x})] = 1 - p$ are equivalent in difficulty, yielding equal optimal entropy: $H^*(p) = H^*(1 - p)$. Combined with the concavity of MaxEnt solutions (from convex optimization theory), this symmetry implies that $H^*(p)$ achieves its unique maximum at $p = 0.5$ and decreases monotonically as $p$ moves toward the extremes 0 or 1. The complete proof appears in Appendix F. □

While our analysis focuses on binary classification, the theoretical challenges in extending these results to multiclass settings and their relationship to our empirical findings are discussed on Appendix F.4.

## 6 CONCLUSION

We introduce Probabilistic Prompting, a framework that achieves fine-grained control over LLM text generation through continuous probability vectors, revealing systematic relationships between prompt entropy and generation fidelity that vary by task dimensionality. Models trained on our synthetic data achieve improved accuracy and calibration, demonstrating that preserving continuous probability structure provides richer supervisory signals than traditional discrete approaches. This work establishes both theoretical foundations and practical tools for more nuanced synthetic data generation that better reflects the continuous nature of semantic properties. Future work extending these principles to multiclass settings and other semantic properties could enable even more sophisticated controllable generation systems. The entropy-diversity correspondence we establish suggests fundamental principles that may apply broadly to neural text generation.

REPRODUCIBILITY STATEMENT

To ensure reproducibility of our results, we provide comprehensive implementation details throughout this work. Section 4.1 specifies all generation hyperparameters, including the use of gemini-2.0-flash with temperature 1.0, Chain-of-Thought prompting, and our principled sampling strategies from Beta(0.5,0.5) and Dirichlet(0.5) distributions. The complete SoftGen algorithm is provided in Algorithm 1 in Appendix A. All prompt templates used for each dataset are included in Appendix C. Downstream classification details are fully specified in Appendix E, including DistilBERT-base-uncased configuration, LoRA hyperparameters, training procedures with AdamW optimizer, and infrastructure details. We conduct all experiments with fixed random seeds (42-46) across five independent trials to ensure statistical reliability. Dataset preprocessing steps and evaluation protocols follow established benchmarks as detailed in Appendix B. Upon acceptance, we will release our complete codebase including generation scripts, training pipelines, and evaluation code to facilitate exact replication of our results.

LLM USAGE STATEMENT

Large Language Models were used to assist with writing and presentation of this manuscript, including improving clarity, refining sentence structure, and proofreading. All research ideas, methodology, experimental design, theoretical contributions, and scientific conclusions are entirely the work of the human authors. LLMs did not contribute to research ideation or the generation of novel scientific insights.

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

# A   PSEUDOCODE

---

**Algorithm 1** The SoftGen Algorithm

---

1: **Input:** Target dataset size $N$, label sampling distribution $\mathcal{P}$, prompt function $\pi$, generator LLM $G$, verifier LLM $\tilde{G}$, discrepancy threshold $\tau$.
2: **Output:** A high-fidelity soft-labeled dataset $\mathcal{D}$.
3:
4: **function** SOFTGEN($N, \mathcal{P}, \pi, G, \tilde{G}, \tau$)
5: $\quad$ $\mathcal{D} \leftarrow \emptyset$

6: $\quad$ $\triangleright$ *Generate samples until the target size is reached*
7: $\quad$ **while** $|\mathcal{D}| < N$ **do**
8: $\quad\quad$ $\triangleright$ Stage 1: Label Sampling
9: $\quad\quad$ $\boldsymbol{p} \sim \mathcal{P}(\cdot)$

10: $\quad\quad$ $\triangleright$ Stage 2: Text Generation
11: $\quad\quad$ $\boldsymbol{x} \sim G(\cdot|\pi(\boldsymbol{p}))$

12: $\quad\quad$ $\triangleright$ Stage 3: Self-Verification
13: $\quad\quad$ $\tilde{\boldsymbol{p}} \leftarrow \tilde{G}(\boldsymbol{x})$
14: $\quad\quad$ $d \leftarrow \|\boldsymbol{p} - \tilde{\boldsymbol{p}}\|_1$

15: $\quad\quad$ **if** $d \leq \tau$ **then**
16: $\quad\quad\quad$ Add $(\boldsymbol{x}, \boldsymbol{p})$ to $\mathcal{D}$
17: $\quad\quad$ **end if**
18: $\quad$ **end while**

19: $\quad$ **return** $\mathcal{D}$
20: **end function**

---

# B   DATASET DETAILS

Table 5: Summary of datasets used in our experiments. Sizes refer to the original benchmark splits.

| Characteristic | **IMDb** | **SST** | **SUBJ** | **Emotion** | **Yahoo! Answers** |
|---|---|---|---|---|---|
| Task Type | Sentiment | Sentiment | Subjectivity | Emotion | Question Category |
| $K$ (No. classes) | 2 | 2 | 2 | 6 | 10 |
| $N_{\text{train}}$ | 25000 | 6920 | 8000 | 89832 | 140000 |
| $N_{\text{test}}$ | 25000 | 1821 | 2000 | 20000 | 60000 |

### IMDB

The IMDb dataset is a large-scale sentiment classification corpus consisting of 50,000 highly polarized movie reviews, equally split into training and testing sets. It is commonly used to evaluate model performance on long-form text classification.

- **Source:** `https://huggingface.co/datasets/stanfordnlp/imdb`
- **Task:** Binary sentiment classification
- **Split:** 25,000 train / 25,000 test
- **Features:** `review`, `label`
- **Label:** 0 = negative, 1 = positive
- **Balance:** Equal class distribution
- **Preprocessing:** Neutral instances removed, reviews pre-cleaned

SST (STANFORD SENTIMENT TREEBANK)

The SST dataset is a fine-grained sentiment analysis corpus that includes fully labeled parse trees, enabling detailed analysis of compositional sentiment in language. Each sentence is annotated with a continuous sentiment score ranging from 0.0 (very negative) to 1.0 (very positive). The dataset also includes sub-sentence (phrase) annotations and syntactic parse trees in both parent-pointer and Penn Treebank formats.

- **Source:** `https://huggingface.co/datasets/stanfordnlp/sst`
- **Task:** Sentiment scoring (regression) and classification
- **Split:** Train / Validation / Test for sentence-level data
- **Features:** `sentence`, `label`, `tokens`, `tree`, `ptb_tree`, `phrase`
- **Label:** Real-valued sentiment score $\in [0.0, 1.0]$
- **Configurations:**
    - *default:* sentence-level regression
    - *dictionary:* phrase-level annotations (no split)
    - *ptb:* full parse trees with discrete labels (0–4)

SUBJ (SUBJECTIVITY CLASSIFICATION)

The SUBJ dataset is a benchmark for classifying whether a sentence expresses a subjective opinion or an objective statement. It is widely used in tasks like sentence representation evaluation.

- **Source:** `https://huggingface.co/datasets/SetFit/subj`
- **Task:** Binary subjectivity classification
- **Split:** 8,000 train / 2,000 test
- **Features:** `text`, `label`, `label_text`
- **Label:** 0 = objective, 1 = subjective
- **Balance:** Balanced between classes

EMOTION

The Emotion dataset contains short English tweets labeled with one of six basic emotions: *sadness*, *joy*, *love*, *anger*, *fear*, and *surprise*. It is used to evaluate fine-grained emotional understanding in social media texts.

- **Source:** `https://huggingface.co/datasets/dair-ai/emotion`
- **Task:** Multi-class emotion classification (6-way)
- **Split:** 16,000 train / 2,000 validation / 2,000 test
- **Unsplit:** 416,809 (14,972 examples for the minority *surprise* class)
- **Features:** `text`, `label`
- **Label:** 0 = sadness, 1 = joy, 2 = love, 3 = anger, 4 = fear, 5 = surprise
- **Balance:** Imbalanced
- **Note**: In our experiments, we used the *unsplit* set as gold training data, and the *split* set as the test set. Specifically, each class is downsampled to the minority class count.

YAHOO! ANSWERS

The Yahoo! Answers dataset is a collection of question-answer pairs for topic classification, constructed from the 10 largest main categories on the Yahoo! Answers platform. The dataset contains question titles, question content, and corresponding best answers selected by the community.

- **Source:** `https://huggingface.co/datasets/yassiracharki/Yahoo_Answers_10_categories_for_NLP`
- **Task:** Multi-class question topic classification (10-way)
- **Split:** 140,000 train / 6,0000 test
- **Features:** `question_title`, `question_content`, `best_answer`, `class`
- **Label:** 0 = Society & Culture, 1 = Science & Mathematics, 2 = Health, 3 = Education & Reference, 4 = Computers & Internet, 5 = Sports, 6 = Business & Finance, 7 = Entertainment & Music, 8 = Family & Relationships, 9 = Politics & Government
- **Balance:** Balanced across categories
- **Note:** The original dataset contains 1,400,000 training instances (140,000 per class), but we downsampled to 10% for computational efficiency while maintaining balanced class distributions. We concatenate `question_title` and `question_content` as the input text, excluding the answer content.

# C PROMPTS

## C.1 TEXT GENERATION PROMPTS

```
prompt = """
    You are an AI assistant that generates realistic movie reviews.
    You will be given a sentiment score from 0.0 to 1.0,
    where 0.0 is negative and 1.0 is positive.
    Guidelines:
    - The sentiment of your review must accurately match the score.
    - Write in a natural and conversational style.
    - Be specific about what you liked or disliked,
      focusing on elements like plot, acting, and direction.
    - Use a varied vocabulary and sentence structure.
    - Avoid repeating the same phrases.
    - The review should be between 100 and 500 words.
    - Avoid overly short or excessively long responses.
    - {chain_of_thought_instructions}

    Generate a movie review with a sentiment score of {score}.
"""
```

Figure 4: Prompt template used to generate text for IMDb

```
prompt = """
    You are an AI assistant generating realistic movie review snippets.
    You will be given a sentiment score from 0.0 to 1.0,
    where 0.0 is negative, and 1.0 is positive.
    Guidelines:
    - The sentiment of your snippet must accurately match the score.
    - Write in a natural and conversational style.
    - Be specific about what you liked or disliked, focusing on
      elements like plot, acting, direction, or overall experience.
    - Use a varied vocabulary and sentence structure.
    - Avoid repeating the same phrases or patterns.
    - The snippet should be between 10 and 300 words.
    - Make it sound like a genuine excerpt from a longer review.
    - {chain_of_thought_instructions}

    Generate a movie review snippet with a sentiment score of {score}.
"""
```

Figure 5: Prompt template used to generate text for SST

```
prompt = """
    You are an AI assistant tasked with generating realistic
    text to train a subjectivity classifier.
    Your goal is to write natural-sounding text that accurately
    reflects a given subjectivity score.
    Use a subjectivity scale from 0.0 to 1.0, where:
    - 0.0 is 'very objective' (a purely factual, neutral statement).
    - 0.5 is 'moderately subjective' (a mix of fact and opinion).
    - 1.0 is 'very subjective' (an entirely personal opinion or feeling).
    Guidelines:
    - The subjectivity level of your text must accurately match the score.
    - Write in a natural, everyday style, like a short social media post.
    - Use a varied vocabulary, sentence structure, and range of topics.
    - Objective text (near 0.0) should state facts, descriptions,
      or neutral information.
    - Subjective text (near 1.0) should contain personal opinions,
      feelings, or judgments.
    - The text should generally be between 20 and 250 words.
    - {chain_of_thought_instructions}

    Generate a text with a subjectivity score of {score}.
"""
```

Figure 6: Prompt template used to generate text for SUBJ

```
prompt = """
    You are an AI language model assistant trained
    to generate emotionally nuanced short messages.
    You will receive an emotion distribution across six emotions
    (sadness, joy, love, anger, fear, surprise),
    where each value indicates the strength of emotion and all sum to 1.
    Guidelines:
    - The emotional distribution of your text must match the scores.
    - Write in a natural and conversational style,
      like real social media posts or messages.
    - Be specific about emotional context and situations
      that would naturally evoke these feelings.
    - Use a varied vocabulary and sentence structure.
      Avoid repeating the same phrases or patterns.
    - The message should be between 10 and 300 words.
    - Make it sound like a genuine personal expression of emotion.
    - {chain_of_thought_instructions}

    Generate a message reflecting the following distribution: {score}.
"""
```

Figure 7: Prompt template used to generate text for Emotion

```
prompt = """
    You are an AI assistant that generates realistic questions
    for an online Q&A community forum to train a topic classifier.
    You will be given a category distribution across ten categories:
    ('Society & Culture', 'Science & Mathematics', 'Health',
     'Education & Reference', 'Computers & Internet', 'Sports',
     'Business & Finance', 'Entertainment & Music',
     'Family & Relationships', 'Politics & Government'), where each
     value indicates the strength of that category in probabilities,
     and all values sum to 1.
    Guidelines:
    - The category distribution of your question must match the scores.
    - Write in a natural conversational style, like real forum questions.
    - Be specific about the situation and context that would
      naturally relate to these categories.
    - Use a varied vocabulary and sentence structure.
      Avoid repeating the same phrases or patterns.
    - The question should be between 20 and 500 words.
    - Make it sound like a genuine question from someone
      seeking help or information.
    - {chain_of_thought_instructions}

    Generate a question reflecting the following distribution: {score}.
"""
```

Figure 8: Prompt template used to generate text for Yahoo! Answers

```
chain_of_thought_instructions = """
    Before writing your final answer, please THINK STEP-BY-STEP:
    1. Understand the task requirements and constraints.
    2. Plan your approach and structure.
    3. Consider how to achieve the specified score naturally.
    4. Think about vocabulary, tone, and style choices.
    5. Ensure your response meets all guidelines.
    Provide the complete step-by-step thinking process in the final output.
"""
```

Figure 9: Zero-shot Chain-of-Thought instructions

## C.2 SELF-VERIFICATION & JUDGE PROMPTS

```
self_verify_or_judge_prompt = """
    You are an expert sentiment analyst. Your task is to evaluate the
    sentiment of given text and provide a score between 0 and 1, where:
    - 0 represents negative sentiment
    - 0.5 represents neutral sentiment
    - 1 represents positive sentiment
    IMPORTANT: Provide your sentiment score with 3 decimal places
    of precision (e.g., 0.123, 0.456, 0.789). Do not round to simple
    fractions like 0.1, 0.2, etc.
    Be thorough in your analysis and provide reasoning for your judgment.
    Please analyze the sentiment of the following text: {text}
    Provide a sentiment score between 0 and 1 with 3 decimal places
    of precision (e.g., 0.123, 0.456, 0.789),
    along with your reasoning and confidence level.
"""
```

Figure 10: Prompt template used to self-verify/judge text generated for IMDb & SST.

```
self_verify_or_judge_prompt = """
    You are an expert in analyzing text subjectivity.
    Your task is to evaluate the subjectivity of given text
    and provide a score between 0 and 1, where:
    - 0 represents objective text (factual description, neutral narration)
    - 0.5 represents moderately subjective text"
    - 1 represents subjective text
      (personal opinions, emotional expressions, evaluative language)
    IMPORTANT: Provide your subjectivity score with 3 decimal places
    of precision (e.g., 0.123, 0.456, 0.789). Do not round to simple
    fractions like 0.1, 0.2, etc.
    Be thorough in your analysis and provide reasoning for your judgment.
    Please analyze the subjectivity of the following text: {text}
    Provide a subjectivity score between 0 and 1 with 3 decimal places
    of precision (e.g., 0.123, 0.456, 0.789),
    along with your reasoning and confidence level.
"""
```

Figure 11: Prompt template used to self-verify/judge text generated for SUBJ.

```
self_verify_or_judge_prompt = """
    You are an expert in analyzing emotional content in text.
    Your task is to evaluate the emotional distribution of given text
    across six emotions: [sadness, joy, love, anger, fear, and surprise].
    For each emotion, provide a score between 0 and 1 indicating
    the strength of that emotion in the text.
    All six scores should sum to 1.0.
    IMPORTANT: Provide each emotion score with 3 decimal places
    of precision (e.g., 0.123, 0.456, 0.789). Do not round to simple
    fractions like 0.1, 0.2, etc.
    Be thorough in your analysis and provide reasoning for your judgment.
    Please analyze the emotional content of the following text: {text}
    Provide an emotion distribution across six emotions
    (sadness, joy, love, anger, fear, surprise) where
    each value is between 0 and 1 and all values sum to 1,
    along with your reasoning and confidence level.
"""
```

Figure 12: Prompt template used to self-verify/judge text generated for Emotion.

```
self_verify_or_judge_prompt = """
    You are an expert in analyzing question categories.
    Your task is to evaluate the category distribution of given questions
    and provide a score between 0 and 1 for each category, where:
    - 0 represents no relevance to the category
    - 1 represents very relevant to the category
    IMPORTANT: Provide each category score with 3 decimal places
    of precision (e.g., 0.123, 0.456, 0.789). Do not round to simple
    fractions like 0.1, 0.2, etc.
    Be thorough in your analysis and provide reasoning for your judgment.
    Please analyze the category distribution of the following: {text}
    Provide a category distribution across ten categories
    (Society & Culture, Science & Mathematics, Health,
    Education & Reference, Computers & Internet, Sports,
    Business & Finance, Entertainment & Music, Family & Relationships,
    Politics & Government) where each value is between 0 and 1 and
    all values sum to 1, along with your reasoning and confidence level.
"""
```

Figure 13: Prompt template used to self-verify/judge text generated for Yahoo! Answers.

## D  SAMPLING DISTRIBUTIONS

Table 6: Ablation study on sampling distribution (Dirichlet $\alpha$) for Emotion. Macro F1 scores for hard vs. soft target training.

| Sampling Distribution | | Hard Target (F1 ↑) | Soft Target (F1 ↑) |
|---|---|---|---|
| **Shape** | $\alpha$ | | |
| Very Flat | 5.0 | 40.9 (0.1) | 44.3 (0.3) |
| Flat | 1.0 | 42.7 (0.4) | 46.9 (0.1) |
| Slightly Spiky | 0.5 | **42.8** (0.3) | **47.1** (0.2) |
| Spikier | 0.3 | 42.4 (0.3) | 46.5 (0.3) |
| Very Spiky | 0.1 | 41.9 (0.1) | 45.5 (0.1) |

# E  DISTILBERT TRAINING DETAILS

## E.1  MODEL ARCHITECTURE AND CONFIGURATION

We use DistilBERT-base-uncased as the base model for all downstream classification experiments. To enable efficient fine-tuning, we employ Low-Rank Adaptation (LoRA) (Hu et al., 2022) with the following configuration:

- **LoRA rank**: $r = 4$
- **Target modules**: Attention layers only (i.e., q_lin and v_lin)
- **LoRA alpha**: $\alpha = 8$
- **LoRA dropout**: 0.1

## E.2  TRAINING HYPERPARAMETERS

All models are trained using the following hyperparameters:

- **Learning rate**: $2 \times 10^{-5}$ with cosine decay schedule
- **Warmup**: Linear warmup for 10% of total training steps
- **Optimizer**: AdamW with weight decay of 0.001
- **Maximum epochs**: 100
- **Batch size**: 128 (unified across all experiments)
- **Mixed precision**: FP16 training for computational efficiency
- **Maximum sequence length**: 512 tokens (DistilBERT's maximum)
- **Text preprocessing**: Texts longer than 512 tokens are truncated

## E.3  LOSS FUNCTION AND TRAINING PROCEDURE

For models trained with soft targets (SoftGen+SoftTarget), we use standard cross-entropy loss applied to the soft label distributions:

$$\mathcal{L} = -\sum_{i=1}^{K} p_i \log(\hat{p}_i) \tag{5}$$

where $p_i$ is the target soft label probability for class $i$ and $\hat{p}_i$ is the model's predicted probability.

For hard target baselines, we use standard cross-entropy loss with one-hot encoded labels.

## E.4  MODEL SELECTION AND VALIDATION

We employ early stopping based on validation performance to prevent overfitting. For each synthetic training dataset, we reserve 10% of the generated samples as a validation set, maintaining the same label distribution as the training set. Training is stopped when validation loss fails to improve for 5 consecutive epochs.

## E.5  INFRASTRUCTURE AND REPRODUCIBILITY

All experiments are conducted on a single NVIDIA RTX 4090 GPU using PyTorch and the Hugging Face Transformers library with FP16 mixed precision training. To ensure reproducibility, we conduct five independent training runs with random seeds 42, 43, 44, 45, and 46. All reported results represent the mean and standard error across these five trials.

We do not apply any dataset-specific class balancing techniques, allowing the natural label distributions from our generation methods to be preserved during training.

# F  COMPLETE PROOF OF BINARY ENTROPY CORRESPONDENCE

This appendix provides the derivation of the exponential family solution and the complete proof of Theorem 1.

## F.1  DERIVATION OF EXPONENTIAL FAMILY SOLUTION

We first establish that the MaxEnt optimization problem yields an exponential family distribution.

**Proposition 1** (Exponential Family Form of MaxEnt Solution). *The solution to the MaxEnt problem in Definition 2 has exponential family form:*

$$P^*(\boldsymbol{x}|\boldsymbol{p}) = \frac{\exp(\boldsymbol{\lambda}^*(\boldsymbol{p})^T \boldsymbol{p}^*(\boldsymbol{x}))}{Z(\boldsymbol{\lambda}^*(\boldsymbol{p}))} \tag{6}$$

*where $\boldsymbol{\lambda}^*(\boldsymbol{p})$ satisfies the constraint equation.*

*Proof.* The MaxEnt optimization problem is:

$$\max_P \quad H(P) = -\sum_{\boldsymbol{x} \in \mathcal{X}} P(\boldsymbol{x}) \log P(\boldsymbol{x}) \tag{7}$$

$$\text{s.t.} \quad \sum_{\boldsymbol{x} \in \mathcal{X}} P(\boldsymbol{x}) \boldsymbol{p}^*(\boldsymbol{x}) = \boldsymbol{p} \tag{8}$$

$$\sum_{\boldsymbol{x} \in \mathcal{X}} P(\boldsymbol{x}) = 1, \quad P(\boldsymbol{x}) \geq 0 \tag{9}$$

We form the Lagrangian:

$$\mathcal{L} = -\sum_{\boldsymbol{x}} P(\boldsymbol{x}) \log P(\boldsymbol{x}) - \boldsymbol{\lambda}^T \left( \sum_{\boldsymbol{x}} P(\boldsymbol{x}) \boldsymbol{p}^*(\boldsymbol{x}) - \boldsymbol{p} \right) - \mu \left( \sum_{\boldsymbol{x}} P(\boldsymbol{x}) - 1 \right) \tag{10}$$

where $\boldsymbol{\lambda} \in \mathbb{R}^K$ and $\mu \in \mathbb{R}$ are Lagrange multipliers.

Taking the derivative with respect to $P(\boldsymbol{x})$ and setting to zero:

$$\frac{\partial \mathcal{L}}{\partial P(\boldsymbol{x})} = -\log P(\boldsymbol{x}) - 1 - \boldsymbol{\lambda}^T \boldsymbol{p}^*(\boldsymbol{x}) - \mu = 0 \tag{11}$$

Solving for $P(\boldsymbol{x})$:

$$P(\boldsymbol{x}) = \exp(-1 - \boldsymbol{\lambda}^T \boldsymbol{p}^*(\boldsymbol{x}) - \mu) \tag{12}$$

Using the normalization constraint $\sum_{\boldsymbol{x}} P(\boldsymbol{x}) = 1$:

$$\sum_{\boldsymbol{x}} \exp(-1 - \boldsymbol{\lambda}^T \boldsymbol{p}^*(\boldsymbol{x}) - \mu) = 1 \tag{13}$$

This gives us:

$$\exp(-1 - \mu) = \frac{1}{\sum_{\boldsymbol{x}} \exp(\boldsymbol{\lambda}^T \boldsymbol{p}^*(\boldsymbol{x}))} \equiv \frac{1}{Z(\boldsymbol{\lambda})} \tag{14}$$

Therefore:

$$P^*(\boldsymbol{x}) = \frac{\exp(\boldsymbol{\lambda}^T \boldsymbol{p}^*(\boldsymbol{x}))}{Z(\boldsymbol{\lambda})} \tag{15}$$

The optimal multiplier $\boldsymbol{\lambda}^*(\boldsymbol{p})$ is determined by the constraint:

$$\mathbb{E}_{P^*}[\boldsymbol{p}^*(\boldsymbol{x})] = \sum_{\boldsymbol{x}} P^*(\boldsymbol{x}) \boldsymbol{p}^*(\boldsymbol{x}) \tag{16}$$

$$= \sum_{\boldsymbol{x}} \frac{\exp(\boldsymbol{\lambda}^T \boldsymbol{p}^*(\boldsymbol{x}))}{Z(\boldsymbol{\lambda})} \boldsymbol{p}^*(\boldsymbol{x}) \tag{17}$$

$$= \frac{\partial \log Z(\boldsymbol{\lambda})}{\partial \boldsymbol{\lambda}} = \boldsymbol{p} \tag{18}$$

This establishes the exponential family form with the constraint equation $\frac{\partial \log Z(\boldsymbol{\lambda})}{\partial \boldsymbol{\lambda}}\big|_{\boldsymbol{\lambda}=\boldsymbol{\lambda}^*(\boldsymbol{p})} = \boldsymbol{p}$.

$\square$

## F.2 PROOF OF BINARY ENTROPY-DIVERSITY CORRESPONDENCE

**Theorem 2** (Binary Entropy-Diversity Correspondence - Restated). *Consider binary classification* ($K = 2$) *under Assumptions 1–3. For prompts* $\boldsymbol{p}_1 = [p_1, 1 - p_1]$ *and* $\boldsymbol{p}_2 = [p_2, 1 - p_2]$ *with* $p_1, p_2 \in (0, 1)$*:*

*If* $|p_1 - 0.5| > |p_2 - 0.5|$ *(i.e.,* $\boldsymbol{p}_1$ *is more extreme than* $\boldsymbol{p}_2$*), then* $H^*(\boldsymbol{p}_1) < H^*(\boldsymbol{p}_2)$*.*

*Proof.* We prove this by showing that $H^*(p)$ is maximized at $p = 0.5$ and decreases monotonically as $p$ moves away from $0.5$.

**Step 1: Binary Exponential Family Setup** From Proposition 1, for binary classification ($K = 2$), the optimal distribution has the form:

$$P^*(\boldsymbol{x}|p) = \frac{\exp(\lambda^*(p) \cdot p_1^*(\boldsymbol{x}))}{Z(\lambda^*(p))} \tag{19}$$

where $p_1^*(\boldsymbol{x}) \in [0, 1]$ is the probability that text $\boldsymbol{x}$ belongs to class 1, and:

$$Z(\lambda) = \sum_{\boldsymbol{x} \in \mathcal{X}} \exp(\lambda \cdot p_1^*(\boldsymbol{x})) \tag{20}$$

**Step 2: Constraint Equation** The calibration constraint $\mathbb{E}_{P^*}[p_1^*(\boldsymbol{x})] = p$ gives us:

$$\frac{\partial \log Z(\lambda)}{\partial \lambda}\bigg|_{\lambda=\lambda^*(p)} = p \tag{21}$$

This implicitly defines $\lambda^*(p)$ as a function of $p$.

**Step 3: Binary Symmetry Properties** For binary classification, Assumption 2 means: for every text $\boldsymbol{x}$ with label $[p, 1 - p]$, there exists a corresponding text $\boldsymbol{x}'$ with the reflected label $[1 - p, p]$.

**Lemma 1** (Binary Symmetry of Optimal Entropy). *For any* $p \in (0, 1)$*:* $H^*(p) = H^*(1 - p)$*.*

*Proof of Lemma.* The reflection symmetry assumption defines a bijection $T : \mathcal{X} \to \mathcal{X}$ where if $\boldsymbol{p}^*(\boldsymbol{x}) = [s, 1 - s]$ then $\boldsymbol{p}^*(T(\boldsymbol{x})) = [1 - s, s]$.

Let $P^*$ be optimal for constraint $\mathbb{E}[p_1^*(\boldsymbol{x})] = p$. Define $Q(\boldsymbol{x}) = P^*(T(\boldsymbol{x}))$.

Then:

$$\mathbb{E}_Q[p_1^*(\boldsymbol{x})] = \sum_{\boldsymbol{x}} Q(\boldsymbol{x}) \cdot p_1^*(\boldsymbol{x}) \tag{22}$$

$$= \sum_{\boldsymbol{x}} P^*(T(\boldsymbol{x})) \cdot p_1^*(\boldsymbol{x}) \tag{23}$$

$$= \sum_{\boldsymbol{y}} P^*(\boldsymbol{y}) \cdot p_1^*(T^{-1}(\boldsymbol{y})) \tag{24}$$

$$= \sum_{\boldsymbol{y}} P^*(\boldsymbol{y}) \cdot (1 - p_1^*(\boldsymbol{y})) \tag{25}$$

$$= 1 - \sum_{\boldsymbol{y}} P^*(\boldsymbol{y}) \cdot p_1^*(\boldsymbol{y}) \tag{26}$$

$$= 1 - p \tag{27}$$

Since $T$ is a bijection: $H(Q) = H(P^*)$.

By optimality: $H^*(1 - p) \geq H(Q) = H(P^*) = H^*(p)$.

By applying the same argument in reverse (since $T^{-1}$ also satisfies the symmetry property): $H^*(p) \geq H^*(1-p)$.

Therefore: $H^*(1-p) = H^*(p)$. □

**Step 4: Critical Point Analysis** From the symmetry property $H^*(p) = H^*(1-p)$, combined with smoothness (Assumption 3), we establish that $p = 0.5$ is a critical point.

Differentiating $H^*(p) = H^*(1-p)$ with respect to $p$:

$$\frac{dH^*}{dp} = \frac{dH^*}{d(1-p)} \cdot \frac{d(1-p)}{dp} = -\frac{dH^*}{dp}\bigg|_{1-p} \tag{28}$$

At $p = 0.5$:

$$\frac{dH^*}{dp}\bigg|_{p=0.5} = -\frac{dH^*}{dp}\bigg|_{p=0.5} \tag{29}$$

This implies $\frac{dH^*}{dp}\bigg|_{p=0.5} = 0$, confirming that $p = 0.5$ is a critical point.

**Step 5: Concavity Analysis**

**Lemma 2** (Concavity of Binary MaxEnt Solution). *The optimal entropy $H^*(p)$ is strictly concave in $p$.*

*Proof of Lemma.* This follows from parametric optimization theory. We are maximizing the concave objective $H(P)$ subject to the linear constraint $\mathbb{E}_P[p_1^*(\boldsymbol{x})] = p$.

The optimal value function of a concave optimization problem is concave in the constraint parameters. Specifically, for the parametric problem:

$$f(p) = \max_P \quad H(P) \tag{30}$$
$$\text{s.t.} \quad \mathbb{E}_P[p_1^*(\boldsymbol{x})] = p \tag{31}$$

Since $H(P)$ is concave in $P$ and the constraint is linear in $P$, the optimal value function $f(p) = H^*(p)$ is concave in the parameter $p$. This is a standard result in convex optimization (see Boyd & Vandenberghe (2004)).

Moreover, the strict concavity follows from the strict concavity of the entropy function and the non-degeneracy of the constraint (ensured by Assumption 3). □

**Step 6: Conclusion** Combining the three key results:

- **Symmetry**: $H^*(p) = H^*(1-p)$ for all $p \in (0, 1)$

- **Critical point**: $\frac{dH^*}{dp}\big|_{p=0.5} = 0$

- **Strict concavity**: $H^*(p)$ is strictly concave in $p$

These three properties together uniquely determine the behavior of $H^*(p)$:

1. **Existence of maximum**: Since $H^*(p)$ is continuous and concave on the compact interval $[0, 1]$, it achieves its maximum.

2. **Uniqueness of maximum**: Since $H^*(p)$ is strictly concave, it has at most one maximum.

3. **Location of maximum**: The symmetry property $H^*(p) = H^*(1-p)$ combined with the critical point condition places the unique maximum at $p = 0.5$.

4. **Monotonic decrease**: Since $H^*(p)$ is strictly concave with a unique maximum at $p = 0.5$, it decreases monotonically as $p$ moves away from $0.5$ in either direction.

Therefore: if $|p_1 - 0.5| > |p_2 - 0.5|$ (meaning $p_1$ is farther from $0.5$ than $p_2$), then $H^*(p_1) < H^*(p_2)$.

This establishes the entropy-diversity correspondence for binary classification: more balanced prompts yield higher entropy outputs. $\qquad\square$

### F.3 THE ROLE OF PROMPT CONSTRUCTION

Assumption 4 is critical for connecting our theoretical framework to practical implementation. This assumption means that $\pi : \Delta^{K-1} \to \mathcal{L}$ (where $\mathcal{L}$ is the space of natural language prompts) is an **ideal information channel**: no information about the constraint is lost or misinterpreted. In practice, this idealization may not hold due to (1) **information loss** in $\pi(\boldsymbol{p})$, (2) **interpretation errors** by the LLM, and (3) **approximation errors** in achieving the MaxEnt solution. Our theoretical results provide a performance upper bound. The empirical validation demonstrates how closely this bound can be approached with well-designed prompt construction.

### F.4 THEORETICAL LIMITATIONS

While we have proven the entropy-diversity correspondence rigorously for binary classification, extending these results to multiclass cases ($K > 2$) faces fundamental theoretical challenges. The multiclass extension would require several strong conditions: (1) **generalized symmetry structure** where the text space exhibits perfect permutation symmetry across all $K$ categories around the uniform distribution $[1/K, \dots, 1/K]$; (2) **preservation of exponential family structure** in the optimal solution; (3) **multi-dimensional concavity** of the optimal value function $H^*(\boldsymbol{p})$; and (4) **entropy maximization at the uniform distribution** to ensure the uniform prompt achieves maximum output diversity.

However, these requirements, particularly the generalized symmetry assumption, are prohibitively strong and conflict with the inherent asymmetries of real semantic spaces. Extending Assumption 2 to require that for any text with label distribution $\boldsymbol{p}^*(\boldsymbol{x})$, there exist corresponding texts with all possible permutations of this distribution is an idealization that rarely holds in practice.

This theoretical limitation is particularly significant given our empirical finding that multiclass tasks exhibit the opposite entropy-error relationship: negative correlations where higher entropy prompts yield better fidelity. We hypothesize that this reversal arises from the generator's systematic underconfidence bias (demonstrated in Section 4.2), which our idealized MaxEnt framework does not capture. In multiclass semantic spaces, this bias manifests as a preference for balanced, moderate outputs across categories. Consequently, low-entropy prompts requesting extreme single-category dominance conflict with the generator's intrinsic tendency toward distributional balance, while high-entropy prompts specifying mixed content align naturally with this conservative behavior.

This suggests that real controllable generators deviate systematically from the idealized MaxEnt solution in ways that depend critically on task dimensionality. While our binary theory provides a performance upper bound and explains the directional relationship between prompt entropy and optimal output diversity, understanding multiclass behavior requires incorporating generator-specific biases that violate our perfect calibration assumptions. Future theoretical work should develop frameworks that account for these systematic deviations from optimality.

