# OpenReview forum: "Continuous Control of LLM Text Generation via Probabilistic Prompting"
_ICLR.cc/2026/Conference — ICLR 2026 Conference Withdrawn Submission_

### Official Review · Reviewer_EvWB · 2025-10-30

**Soundness:** 2
**Presentation:** 2
**Contribution:** 1
**Rating:** 2
**Confidence:** 4

**Summary:**

This paper proposes SoftGen, a zero-shot method that implements probabilistic prompting through sampling probability, text generation, and self-verification. Experiments on generation alignment and downstream task applications are conducted to evaluate its performance.

**Strengths:**

1.The paper is clearly written and easy to follow.

2.Several visual aids are provided to illustrate key ideas and results, effectively supporting the main arguments.

3.Sufficient implementation details are included to enable reproduction of the results.

**Weaknesses:**

1.The novelty of the work is limited. Incorporating probabilities into prompts and using self-verification seem like incremental contribution.

2.Although one of the main focuses of the paper is controlled text generation (CTG), the authors do not discuss relevant prior work in this area.

3.The scope of the study is narrow, as the experiments focus only on text classification. Additional tasks such as commonsense reasoning, truthful text generation, and instruction following should be explored to demonstrate broader applicability.

4.The experiments rely solely on Gemini-2.0-Flash as the backbone large language model (LLM). To ensure generalizability, experiments with other LLMs should be included.

5.The comparison section could be strengthened by including more recent baselines and methods discussed in the related work section.

**Questions:**

N.A.

---

### Official Review · Reviewer_U4zz · 2025-10-31

**Soundness:** 2
**Presentation:** 3
**Contribution:** 2
**Rating:** 4
**Confidence:** 4

**Summary:**

The paper introduces Probabilistic Prompting: conditioning LLM generation on a continuous probability vector (“soft label”) rather than a discrete category. It instantiates this as SoftGen, a zero-shot, three-stage pipeline: (1) sample a soft label from Beta/Dirichlet; (2) prompt an LLM with a natural-language rendering of that vector to generate text; (3) self-verify with an independent LLM instance and accept the sample only if the verified soft label is within a threshold.

They evaluate on five text-classification benchmarks (IMDb, SST, SUBJ, Emotion, Yahoo!). The paper first measures fidelity then for downstream utility by training DistilBERT+LoRA on the synthetic data (hard vs soft targets). They report: (i) high directional alignment between prompts and judges; (ii) systematic relations between prompt entropy and generation error; and (iii) consistent downstream gains over “HardGen” categorical prompting baselines

**Strengths:**

* The paper is presented with a clear formulation and pipeline. The $(S, P, \pi)$ framework and SoftGen’s 3-stage pipeline are well specified and formalized; The verification and acceptance rule are explicitly demonstrated.
* The proposed method has solid downstream improvements. Training on SoftGen data beats hard-label baselines across datasets; soft-target training further improves accuracy/calibration on most tasks.

**Weaknesses:**

* The framework is demonstrated exclusively on classification tasks. Adding generation tasks for experiment with multi-attributes conditioning would strengthen the case. (see CTRL[1], GeDi[2], FAST[3], SteerLM[4] for reference)
* To establish synthetic data as a driver of downstream quality, the paper needs more analysis and studies on effects of scaling and mixing data:
(i) why there's a gap between synthetic data and human-labelled data (Gold), as shown in Table 2; (ii) if scaling the size of synthetic data could help further the downstream model quality; (iii) if mixture of synthetic data and human-labelled data could help improve downstream model accuracy
* Downstream results focus on DistilBERT. Ablations on larger encoders (e.g., BERT-large, RoBERTa-large) and a small LLM/SLM would strengthen generality claims.

[1] Keskar, Nitish Shirish, Bryan McCann, Lav R. Varshney, Caiming Xiong, and Richard Socher. "Ctrl: A conditional transformer language model for controllable generation." arXiv preprint arXiv:1909.05858 (2019).

[2] Krause, Ben, Akhilesh Deepak Gotmare, Bryan McCann, Nitish Shirish Keskar, Shafiq Joty, Richard Socher, and Nazneen Fatema Rajani. "Gedi: Generative discriminator guided sequence generation." arXiv preprint arXiv:2009.06367 (2020).

[3] Chai, Junyi, Reid Pryzant, Victor Ye Dong, Konstantin Golobokov, Chenguang Zhu, and Yi Liu. "Fast: Improving controllability for text generation with feedback aware self-training." arXiv preprint arXiv:2210.03167 (2022).

[4] Dong, Yi, Zhilin Wang, Makesh Narsimhan Sreedhar, Xianchao Wu, and Oleksii Kuchaiev. "Steerlm: Attribute conditioned sft as an (user-steerable) alternative to rlhf." arXiv preprint arXiv:2310.05344 (2023).

**Questions:**

* For (3) Self-Verification, what fraction of generated samples passes the $\tau$ filter per dataset, what's the impact of $\tau$ to data quality and downstream model performance?
* Currently only gemini-2.0-flash is used as generation model for synthetic data, did you have ablation on prompting larger model to measure the model improvement change?
* What changes are required under current theory and experiment framework to extend the method to multiclass cases?

---

### Official Review · Reviewer_Wt4A · 2025-11-01

**Soundness:** 3
**Presentation:** 2
**Contribution:** 2
**Rating:** 2
**Confidence:** 3

**Summary:**

The authors propose a method to get more diverse LM outputs as synthetic to train natural language classifiers on by directly prompting the model to output examples with mixed labels, e.g., a review that is 70% positive. They show that LLMs are surprisingly able to fulfill these requests. They offer a theoretical explanation for why the outputs generated this way have greater diversity.

**Strengths:**

The method is simple. The evaluations are thoughtful at establishing the properties that they claim the outputs have, e.g., using a panel of judge models as well as text classifiers to verify the proportion of each label.

**Weaknesses:**

I felt that the theoretical analysis was a bit difficult to follow. I could not tell if the result was trivial (more diverse label mixtures means more diverse text because otherwise the text with mixed labels would be excluded?) or saying something deeper. I think some of the confusion arises from what entropy is referring here. Is it text entropy? Label entropy? Both?

Overall, my excitement is somewhat muted. My takeaway is that "language models can do this thing that we didn't know they could do", specifically they can generate calibrated text that fits some label mixture. Other than that, it seems that it was already known that classifiers benefit from training on more diverse data and ambiguous examples.

**Questions:**

I would be willing to raise my score somewhat depending on clarification of the theory.

---

### Note · Authors · 2025-12-05

**Comment:**

We thank the reviewers for their thoughtful comments and the area chairs for their time. While we were unable to address all suggestions within the current review cycle, the feedback has given us valuable directions for strengthening the work. We are withdrawing to further develop the paper.

**Withdrawal Confirmation:**

I have read and agree with the venue's withdrawal policy on behalf of myself and my co-authors.